# An Observational Prospective Cohort Study of Incidence and Outcome of *Streptococcus pneumoniae* and *Hemophilus influenzae* Infections in Adult Solid Organ Transplant Recipients

**DOI:** 10.3390/microorganisms9071371

**Published:** 2021-06-24

**Authors:** Omid Rezahosseini, Dina Leth Møller, Søren Schwartz Sørensen, Michael Perch, Finn Gustafsson, Marco Gelpi, Jenny Knudsen, Marie Helleberg, Allan Rasmussen, Susanne Dam Nielsen, Zitta Barrella Harboe

**Affiliations:** 1Viro-Immunology Research Unit, Department of Infectious Diseases, Rigshospitalet, University of Copenhagen, 2100 Copenhagen, Denmark; omid.rezahosseini@regionh.dk (O.R.); dina.leth.moeller@regionh.dk (D.L.M.); marco.gelpi@regionh.dk (M.G.); sdn@dadlnet.dk (S.D.N.); 2Department of Nephrology, Rigshospitalet, University of Copenhagen, 2100 Copenhagen, Denmark; Soeren.Schwartz.Soerensen@regionh.dk; 3Department of Clinical Medicine, University of Copenhagen, 2100 Copenhagen, Denmark; Michael.Perch@regionh.dk (M.P.); Finn.Gustafsson@regionh.dk (F.G.); 4Section for Lung Transplantation, Department of Cardiology, Rigshospitalet, University of Copenhagen, 2100 Copenhagen, Denmark; 5Department of Cardiology, Rigshospitalet, University of Copenhagen, 2100 Copenhagen, Denmark; 6Department of Clinical Microbiology, Rigshospitalet, University of Copenhagen, 2100 Copenhagen, Denmark; Inge.Jenny.Dahl.Knudsen@regionh.dk; 7Centre of Excellence for Health, Immunity and Infections, Department of Infectious Diseases, Rigshospitalet, University of Copenhagen, 2100 Copenhagen, Denmark; Marie.Helleberg@regionh.dk; 8Department of Surgical Gastroenterology and Transplantation, Rigshospitalet, University of Copenhagen, 2100 Copenhagen, Denmark; Allan.Rasmussen@dadlnet.dk; 9Department of Bacteria, Parasites and Fungi, Statens Serum Institut, 2100 Copenhagen, Denmark; 10Department of Pulmonary and Infectious Diseases, Hospital of Nordsjælland, University of Copenhagen, 2100 Copenhagen, Denmark

**Keywords:** *Streptococcus pneumoniae*, invasive pneumococcal diseases, *Hemophilus influenzae*, organ transplant, incidence, hospitalization, mortality

## Abstract

Background: *Streptococcus pneumoniae (S. pneumoniae)* and *Hemophilus influenzae (H. influenzae)* are among the main vaccine-preventable bacterial infections in immunocompromised individuals including solid organ transplant (SOT) recipients. There is a lack of information about incidence and outcomes of these infections in SOT recipients. Methods: We determined the incidence of *S. pneumoniae* and *H. influenzae*, the related hospitalization, and 30- and 180-days mortality in a large cohort of 1182 adult SOT recipients. We calculated 95% confidence intervals (CI) of incidence rate (IR) using Byar’s approximation to the Poisson distribution. Results: The overall IR of *S. pneumoniae* and *H. influenzae* were 1086 (95% CI, 796–1448) and 1293 (95% CI, 974–1687) per 100,000 person-years of follow-up (PYFU), respectively. The IR of invasive infections were 76 (95% CI, 21–202) and 25 (95% CI, 2.3–118) per 100,000 PYFU, respectively. Hospital admission was required in >50%, 30-days mortality was 0, and 180-days mortality was 8.8% and 4.5% after *S. pneumoniae* and *H. influenzae* infections, respectively. Conclusions: The IR of invasive *S. pneumoniae* and *H. influenzae* infections in SOT recipients were much higher than reports from the general population in Denmark. Furthermore, a large proportion of infected SOT recipients were hospitalized. These findings highlight the need for further studies to assess uptake and immunogenicity of vaccines against *S. pneumoniae* and *H. influenzae* in SOT recipients.

## 1. Introduction

Solid organ transplant (SOT) recipients receive life-long immunosuppression and therefore remain susceptible to infections that, in turn, lead to increased morbidity and mortality [1,2]. Etiology and incidence of infections vary according to the type of transplanted organ, time post-transplantation, and immunosuppression [3]. *Streptococcus pneumoniae* (*S. pneumoniae*) and *Hemophilus influenzae* (*H. influenzae*) are common bacterial causes of upper and lower respiratory tract infections and among the main vaccine-preventable bacterial infections post-transplantation [4,5]. Invasive infections, i.e., infections in normally sterile sites, are the most serious manifestations of infections caused by *S. pneumoniae* and *H. influenzae* with mortality rates as high as 29% in SOT recipients [6].

Studies reporting incidence rates (IR) of invasive and non-invasive *S. pneumoniae* and *H. influenzae* infections in SOT recipients are few [4]. However, vaccination against these bacteria is recommended based on data from studies conducted in pediatric SOT recipients, studies conducted prior to initiation of universal *S. pneumoniae* and *H. influenzae* vaccination programs, and studies in either immunocompetent or immunocompromised patients other than SOT recipients [4,5,7].

The aim of this study is to determine the incidence of *S. pneumoniae* and *H. influenzae* infections (invasive and non-invasive), the related hospitalization, and 30- and 180-days all-cause mortality in adult SOT recipients.

## 2. Materials and Methods

### 2.1. Study Design and Participants

In this observational prospective cohort study, we included adult (≥18 years) SOT recipients (heart-, lung-, kidney-, liver-, simultaneous liver and kidney, and simultaneous kidney and pancreas recipients) who underwent SOT at Copenhagen University Hospital, Rigshospitalet, between 1 January 2011 and 31 December 2017. Due to the small number, combined liver and kidney transplant recipients were included in the group of liver transplant recipients. Combined pancreas and kidney transplant recipients were included as a separate group. Rigshospitalet is a highly specialized hospital that performs transplantations and the only center for liver and lung transplantation in Denmark. All SOT recipients were included in the Management of Posttransplant Infections in Collaborating Hospitals (MATCH) cohort. MATCH was established at Rigshospitalet in 2011 to improve management of infections post-transplantation [8].

We obtained clinical characteristics and microbiology data from the Centre of Excellence for Personalized Medicine of Infectious Complications in Immune Deficiency (PERSIMUNE) data warehouse [8]. Data were prospectively collected and merged as part of the routine care. Data from several clinical databases and national registries, such as the national Danish Microbiology Database (MiBa), are combined in the PERSIMUNE data repository. MiBa contains all microbiological data in Denmark from both hospitals and general practice since 2010 [9]. Data on *S. pneumoniae* and *H. influenzae* B vaccinations were not available.

The study was conducted in accordance with the declaration of Helsinki. The National Committee on Health Research Ethics (H-170024315) and the Data Protection Agency (04433, RH-2016-47) approved the retrieval of the data for the entire study. All experimental protocols were approved by the National Committee on Health Research Ethics of Denmark and the Danish Data Protection Agency. All relevant approval for this project was obtained from the Danish Health and Medicine Authorities. Informed consent for this type of study is not required according to national legislations. The National Committee on Health Research Ethics of Denmark waived the need for informed consent. All data was retrieved anonymously from the PERSIMUNE data repository and we did not have access to the medical records during the study therefore no further permissions were required. All methods were performed in accordance with the relevant guidelines and regulations.

### 2.2. Definitions

Invasive infections were defined as positive cultures of *S. pneumoniae* and *H. influenzae* from normally sterile sites (blood, cerebrospinal fluid, synovial fluid, peritoneal fluid, and pleural fluid). Non-invasive infections were defined as positive cultures from non-sterile sites.

Among non-invasive infections, lower respiratory tract infections were defined as a positive cultures of *S. pneumoniae* and *H. influenzae* in bronchoalveolar lavage (BAL) or standard sputum specimens.

### 2.3. Cultures

All microbiological specimens were cultured according to the standard of care. The strain of *H. influenzae* was determined using matrix-assisted laser desorption ionization-time of flight (MALDI-TOF) technique.

An episode of infection was defined as a positive culture from any site. In case of repeated positive cultures for a unique SOT recipient, we considered it as a new episode of infection, if the infection occurred more than 14 days from the previous episode of infection or if the bacterium was different [10].

### 2.4. Follow-Up

SOT recipients were followed from the date of transplantation to a positive culture, retransplantation, death, end of fifth year (day 1826) post-transplantation, or 31 December 2018 whichever came first. Last date of inclusion was 31 December 2017 to allow a period of follow-up for all SOT recipients. Infections that occurred after five years post-transplantation were not reported.

### 2.5. Incidence of S. pneumoniae and H. influenzae Infections

Number of cases and the IR of *S. pneumoniae* and *H. influenzae* infections (invasive and non-invasive) were reported for the entire follow-up period. We reported the overall IR of *S. pneumoniae* and *H. influenzae* infections, the IR of invasive infections, and the IR according to the type of transplanted organ.

### 2.6. Statistical Analysis

Proportions were reported as percentage, and continuous data were reported as medians with interquartile ranges (IQR). Mann–Whitney U test was used to compare the differences in medians, and Fisher’s exact test was used to test the frequency distributions. We calculated 95% confidence intervals (CI) of IR using Byar’s approximation to the Poisson distribution. Estimates of the cumulative incidence of the first episode of *S. pneumoniae* and *H. influenzae* infection were calculated using Aalen-Johansen estimator with death and retransplantation as competing risks. Statistical differences were tested using Grey’s test. In two explorative analysis, we conducted a matched case-control study to compare all-cause mortality at any time after transplantation in SOT recipients who had and did not have *S. pneumoniae* or *H. influenzae* infection. SOT recipients with positive infection were assumed as cases whereas controls were selected from SOT recipients who did not have infection. Cases and controls matched on age, gender, the type of transplantation, and days post-transplantation. Fisher’s exact test was used to test all-cause mortality between cases and controls. MatchIt package was used for the case-control part using the nearest neighbor-matching method and 1:3 ratio. R software version 3.5.2 was used for statistical analyses, and a *p* value ≤ 0.05 was considered statistically significant.

## 3. Results

### 3.1. Patient Characteristics

We included 1182 adults SOT recipients; 577 (49%) kidney, 293 (25%) liver, 210 (18%) lung, 84 (7.1%) heart, and 18 (1.5%) simultaneous pancreas and kidney transplant recipients. Among these, 726 (61%) were male and median age at the time of transplantation was 50 (IQR 41–59) years.

Applying criteria for follow-up that we mentioned in the methods, 24 (2%) SOT recipients were re-transplanted and 169 (14%) died. The median time to retransplantation was 223 (IQR 94–667) days and for the death was 445 (IQR 170–938) days post-transplantation.

SOT recipients had a median of 23 (IQR 8–54) cultures from any site and for any reason. Only 48 (4%) of 1182 SOT recipients did not have any cultures performed during follow-up.

### 3.2. S. pneumoniae Infection

Forty-three episodes of *S. pneumoniae* infections were found in 34 SOT recipients. Four of 34 (12%) SOT recipients had more than one episode. Total of 3 (7%), 35 (81%), and 5 (12%) of 43 episodes were found in blood, lower respiratory tract, and other sites, respectively. The three blood stream infections were the only invasive infections. Patient characteristics of SOT recipients with and without at least one episode of *S. pneumoniae* infection are shown in Table 1. All three SOT recipients with invasive *S. pneumoniae* infection and 15 (48%) of 31 SOT recipients with non-invasive *S. pneumoniae* infection were admitted to hospital. None (0%) and 3 (8.8%) of SOT recipients with *S. pneumoniae* infection died (all-cause) during 30-days and 180-days of follow-up, respectively. In a matched case-control, we did not observe statistically significant difference between all-cause mortality at any time after transplantation in SOT recipients who had and did not have *S. pneumoniae* infection (Appendix A).

### 3.3. Incidence Rates of S. pneumoniae Infection

In analyses of *S. pneumoniae* infections there were 3961 person-years of follow-up (PYFU). The overall IR of *S. pneumoniae* infection (invasive and non-invasive) during the first five years post-transplantation was 1086 per 100,000 PYFU (95% CI, 796–1448). The IR of invasive *S. pneumoniae* infection was 76 per 100,000 PYFU (95% CI, 21–202) (Table 2).

The overall IRs of *S. pneumoniae* infection according to type of transplanted organ are shown in Table 2.

The cumulative incidence of the first episode of *S. pneumoniae* infection in the first five years post-transplantation was 3.9% (95% CI, 0.63–7.2). There were no significant differences in cumulative incidence of *S. pneumoniae* infection according to the type of transplanted organ (*p* = 0.82) (Figure 1A).

### 3.4. H. influenzae Infection

There were 51 episodes of *H. influenzae* infection in 44 SOT recipients. Three (6.8%) of 44 SOT recipients had more than one episode. Total of 1 (2%), 45 (88%), and 5 (10%) of 51 episodes were found in blood, lower respiratory tract, and other sites, respectively. We found three SOT recipients who had simultaneous positive cultures for both *H. influenzae* and *S. pneumoniae* from lower respiratory tract. Patient characteristics of SOT recipients with and without at least one episode of *H. influenzae* infection are shown in Table 3. The SOT recipient with invasive *H. influenzae* infection was not admitted to hospital, but 22 of 43 (51%) SOT recipients with non-invasive *H. influenzae* infection were admitted to hospital. None of 44 SOT recipients with *H. influenzae* infection died during 30-days, but 2 (4.5%) died (all-cause) during 180-days of follow-up after *H. influenzae* infection, none of them had invasive infection. In a matched case-control, we did not observe statistically significant difference between all-cause mortality at any time after transplantation in SOT recipients who had and did not have *H. influenzae* infection (Appendix A).

### 3.5. Incidence Rates of H. influenzae Infection

In analyses of *H. influenzae* infections there were 3941 PYFU. The overall IR of *H. influenzae* infection (invasive and non-invasive) during the first five years post-transplantation was 1293 per 100,000 PYFU (95% CI, 974–1687). The IR of invasive *H. influenzae* infection was 25 per 100,000 PYFU (95% CI, 2–118) (Table 2).

The overall IRs of *H. influenzae* infection according to the type of transplanted organ are shown in Table 2. The cumulative incidence of *H. Influenzae* infection in the first five years post-transplantation was 5.0% (95% CI, 1.8–8.1). There were no significant differences in cumulative incidence of *H. influenzae* infection according to the type of transplanted organ (*p* = 0.43) (Figure 1B).

## 4. Discussion

In this large, prospective study of adult SOT recipients with complete follow-up and nationwide data on all microbiology from both hospitals and general practice, we found that both the overall IR of *S. pneumoniae* and *H. influenzae* infections and IR of invasive infections were considerable. All-cause mortality was, 8.8% and 4.5% during 180 days of follow-up after *S. pneumoniae* and *H. influenzae* infections, respectively.

In a cohort study that was done between 2001 and 2014, the IR of IPD was 414 per 100,000 PYFU [11]. In a recent meta-analysis, the pooled IR of invasive *S. pneumoniae* infections in SOT recipients was 465 per 100,000 PYFU [11] which is about six times higher than our estimates. These studies also included patients from before 2007 when vaccination against *S. pneumoniae* was not yet included in the universal childhood vaccination programs which may explain the discrepancy [11,12]. The 7-valent pneumococcal conjugated vaccine was introduced in Europe in 2001 and in the Danish Childhood Vaccination Program in 2007 [13,14]. Moreover, from October 2012, a limited subsidy has been introduced by the Danish Health and Medicines Authority for pneumococcal vaccination in high-risk adults [15]. The IR of invasive *S. pneumoniae* infections in the 38 to 60 years old general population in Denmark decreased from 14 per 100,000 person-years to 10 per 100,000 PYFU during the period 2011 to 2017, possibly as a result of herd immunity and sporadic vaccinations in adults (Unpublished data from Statens Serum Institut (SSI)). It is likely that SOT recipients in Denmark also have benefitted from this indirect effect of childhood vaccination or sporadically got vaccinated. Furthermore, although a routine program for vaccination of all SOT candidates for liver, lung, heart, and pancreas only was introduced in 2020, it is likely that some of the SOT recipients in our study were vaccinated against *S. pneumoniae* prior to or after transplantation. However, a recent study from Denmark showed that the coverage of vaccination against *S. pneumoniae* in the kidney transplant recipients was only 4% [16]. Moreover, according to unpublished data from Knowledge Center of Transplantation at Rigshospitalet, only 51 (18%) and 2 (0.7%) out of 281 liver transplant recipients who were transplanted from 1 January 2011 to 31 December 2017 had a documented positive history of vaccination against *S. pneumoniae* and *H. influenzae* B, respectively (Unpublished data).

We found a subpopulation of culture positive SOT recipients who had more than one episode of infection with *S. pneumoniae* (12%) and *H. influenzae* (6.8%). It is possible that this subpopulation of SOT recipients received more intensive immunosuppression, but we did not have access to the medication history to confirm this.

The IR of invasive *H. influenzae* infection in adult SOT recipients has not previously been reported. We found the IR of invasive *H. Influenzae* infection was 25 per 100,000 PYFU. For comparison, the mean IR of invasive *H. influenzae* was 1.3 (95% CI 1.1 to 1.5) per 100,000 PYFU in the 38 to 60 years old general population of Denmark in the period 2011–2017 (unpublished data from SSI).

All three SOT recipients with invasive *S. pneumoniae* infection as well as half of the recipients with either non-invasive *S. pneumoniae* or *H. influenzae* infections were admitted to the hospital. The in-hospital mortality after invasive *S. pneumoniae* infection in SOT recipients has previously been reported to be as high as 29% in a study from Canada [6] and the overall mortality in general population within the 28 days after invasive *H. influenzae* infection was 9.8% in a study from Sweden [17]. In our cohort there were no deaths within 30-days, but 8.8% and 4.5% of SOT recipients died from all causes during 180-days after *S. pneumoniae* and *H. influenzae* infections, respectively. Vaccination and herd immunity against *S. pneumoniae* infection, higher awareness about invasive infections and rapid initiation of antibiotics could be the possible explanations for this low mortality. Although, in matched case-control analysis, mortality at any time after transplantation was not different between SOT recipients who had and did not have infection.

The large sample size with complete follow-up, prospective gathering of data, and nationwide data on microbiological tests were the strengths of our study. However, we did not have access to serotype of isolates, immunosuppressive drugs, or vaccination history. Thus, we are unable to determine the effect of these parameters on the incidence of *S. pneumoniae* and *H. influenzae* infections. Moreover, only a few liver and kidney transplant recipients were routinely vaccinated prior to 2020. As such, our data may overestimate the magnitude of infections in a fully vaccinated cohort. In Denmark, the *H. influenzae* is always identified using MALDI-TOF. The *S. pneumoniae* can be identified using MALDI-TOF, but in most cases is identified using the optochin test (optochin susceptible and catalase negative). Only *H. influenzae* type B is vaccine preventable and most of the cases of community-acquired pneumonia, otitis media, and exacerbations of chronic obstructive pulmonary disease are caused by non-typable (i.e., non-vaccine preventable) strains of *H. influenzae* [18]. Unfortunately, we were not able to provide information about the strains of the *S. pneumoniae* and *H. influenzae* which is a limitation to the study.

## 5. Conclusions

The IR of invasive *S. pneumoniae* and *H. influenzae* infections in SOT recipients is much higher than that reported in the general population in Denmark. However, mortality of *S. pneumoniae* and *H. influenzae* infections was lower than expected, albeit a large proportion of infected SOT recipients are hospitalized. *S. pneumoniae* and *H. influenzae* are vaccine preventable diseases, and further studies to assess the uptake and immunogenicity of vaccines against *S. pneumoniae* and *H. influenzae* in SOT recipients are warranted.

## Figures and Tables

**Figure 1 microorganisms-09-01371-f001:**
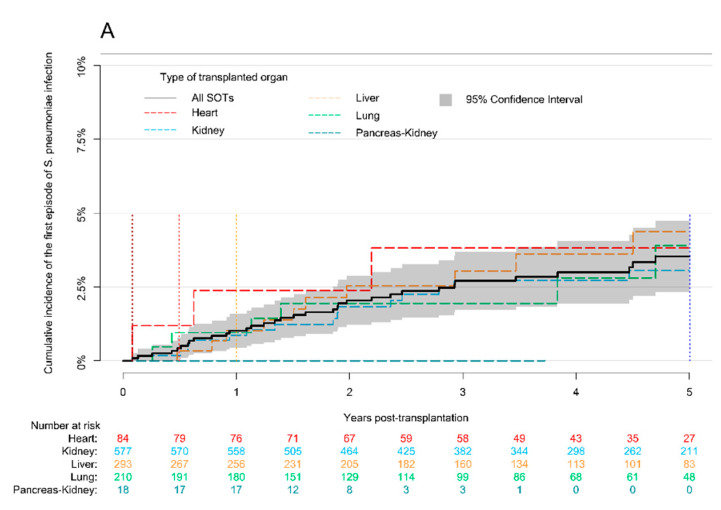
Cumulative incidence of the first episode of *S. pneumoniae* and *H. influenzae* infections during the five years post-transplantation. (**A**) Cumulative incidence (3.9% (95% CI, 0.63–7.2)) of the first episode of *S. pneumoniae* infection. There was no statistically significant difference in cumulative incidence of the first episode of *S. pneumoniae* infection between different types of transplanted organs (*p* = 0.82). (**B**) Cumulative incidence (5.0% (95% CI, 1.8–8.1)) of the first episode of *H. influenzae* infection. There was no statistically significant difference in cumulative incidence of the first episode of *H. influenzae* infection between different types of transplanted organs (*p* = 0.57). Simultaneous kidney and pancreas transplant recipients had no episode of *H. influenzae* infection. Black continuous line shows the cumulative incidence of the first episode of infection in the total cohort of solid organ transplant (SOT) recipients. Colorful dashed lines indicate cumulative incidence of the first episode of infection according to the type of transplanted organ. Vertical dotted-lines determine 30 days (brown), 6 months (red), 1 year (yellow), and 5 years (blue) post-transplantation.

**Table 1 microorganisms-09-01371-t001:** Characteristics of solid organ transplant recipients who had and did not have at least one episode of *S. pneumoniae* infection.

Characteristics		At Least One Positive Culture for *S. pneumoniae* (n = 34)	No Positive Culture for *S. pneumoniae* (n = 1148)	Total (n = 1182)	*p*-Value
Age, median (IQR)		51 (38, 58)	50 (41, 59)	50 (41, 59)	0.63
Male (n, %)		22 (65)	704 (61)	726 (61)	0.73
Transplanted organ (n, %)	Heart	3 (8.8)	81 (7.1)	84 (7.1)	0.90
	Liver	10 (29)	283 (25)	293 (25)	
	Lung	6 (18)	204 (18)	210 (18)	
	Kidney	15 (44)	562 (49)	577 (49)	
	Simultaneous pancreas/kidney	0 (0.0)	18 (1.6)	18 (1.5)	

**Table 2 microorganisms-09-01371-t002:** Incidence rates of *S. pneumoniae* and *H. influenzae* infections.

Category (*N* = Population Size)	Time Interval Post-Transplantation	Incidence Rate (95% CI) of *S. pneumoniae* per 100,000 PYFU	Incidence Rate (95% CI) of *H. influenzae* per 100,000 PYFU
Overall (invasive and non-invasive infections) *N* = 1182	5 years	1086 (796–1448)	1293 (974–1687)
	First year	1235 (707–2017)	1059 (578–1794)
	First six months	873 (331–1914)	1049 (436–2163)
	First month	1000 (91–4662)	1002 (91–4672)
Only invasive infection *N* = 1182	5 years	76 (21–202)	25 (2–118)
	First year	88 (8–411)	88 (8–411)
Heart transplant recipients *N* = 84	5 years	1682 (638–3688)	33 (3–156)
Lung transplant recipients *N* = 210	5 years	996 (414–2053)	1517 (749–2770)
Liver transplant recipients *N* = 293	5 years	1410 (790–2343)	1091 (560–1935)
	Only invasive infection, 5 years	217 (43–695)	109 (10–509)
Kidney transplant recipients *N* = 577	5 years	903 (562–1381)	1477 (1023–2069)
	Only invasive infection, 5 years	48 (4–221)	- *
Simultaneous pancreas and kidney transplant recipients *N* = 18	5 years	- *	- *

* No infection was reported in this group.

**Table 3 microorganisms-09-01371-t003:** Characteristics of solid organ transplant recipients who had and did not have at least one episode of *H. influenzae* infection.

Characteristics		At Least One Positive Culture for *H. influenzae* (n = 44)	No Positive Culture for *H. influenzae* (n = 1138)	Total (n = 1182)	*p*-Value
Age, median (IQR)		55 (46, 60)	50 (41, 59)	50 (41, 59)	0.10
Male (n, %)		26 (59)	700 (62)	726 (61)	0.75
Transplanted organ (n, %)	Heart	1 (2.3)	83 (7.3)	84 (7.1)	0.73
	Liver	10 (23)	283 (25)	293 (25)	
	Lung	9 (21)	201 (18)	210 (18)	
	Kidney	24 (55)	553 (49)	577 (49)	
	Simultaneous pancreas/kidney	0 (0.0)	18 (1.6)	18 (1.5)	

## Data Availability

The data presented in this study are available on request from the corresponding author. The data are not publicly available due to Danish legislations.

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
