# Peer review of "An Observational Prospective Cohort Study of Incidence and Outcome of Streptococcus pneumoniae and Hemophilus influenzae Infections in Adult Solid Organ Transplant Recipients"

_microorganisms, 2021, doi:10.3390/microorganisms9071371_

Round 1

Reviewer 1 Report

Thank you for the opportunity to revise the manuscript entitled: “Incidence and outcome of Streptococcus pneumoniae and Hemophilus influenzae infections in adult solid organ transplant recipients” submitted by Rezahosseini et al. The manuscript highlights the incidence and consequences of the afore-mentioned infections in an immunocompromised population. This is necessarily required as a starting point to further assessments of immunogenicity of vaccines against S. pneumoniae and H. Influenzae infections.

General comments:

Data on S. pneumoniae and H. influenzae B vaccinations were not available in this study population. It can represent a major confounding factor that can significantly compromise the relevance of this manuscript.

Specific comments:

The title and abstract do not include information about the study’s design. It is not present in the methods section too, is this a “pure” prospective observational study?

In the abstract the “Background” section can be completed with a linking sentence which highlights the lack of incidence and outcome information in actual scientific literature.

The keywords do not include the outcomes that the study is aiming to assess (consider adding “hospitalization” and “mortality”). “Organ transplant” can be replaced by “organ transplantation” or “transplants” which are Mesh Terms.

Sub-paragraph 2.4 named “incidence of S. pneumoniae and H. influenzae infections” gives for the first-time general information about the follow-up design, there should be a different sub-paragraph about follow-up to help readers finding this information.

Figure 1 anticipates H. influenzae related results, which are presented to the reader in the next sub-paragraphs. Consider locating this figure after sub-paragraph 3.5.

Lines 112 and 132, information on follow-up duration is missing. Please add, or as suggested before make study design clearer.

In order to improve the quality and relevance of this work, a comparison with at least historical cohort comparison group.

Moreover, having the information regarding immunization status of the participants is extremely relevant, particularly considering the interpretation and conclusion draw by Authors. In light of this, Authors should make some efforts in retrieving this type of information. As for instance call interview? Collecting data from national immunization information register? Other?

The originality and scientific soundness is strictly related to the ability of the Authors to assess immunization status. Otherwise, what is the originality of this work compared to the previous?

Reviewer 2 Report

Overall, this is straightforward study analyzing the incidence of S. pneumoniae and H. influenzae infections in recipients of a variety of solid organ transplants. Some minor issues that require authors' attention are listed below:

  1. Why is the identity of strain types of Sp and Hi not mentioned? This should be an easy addition especially given the use of MALDI-TOF for strain identification.  
  2. Did any SOT recipients develop dual infection with Sp and Hi
  3. It is understood that the data on the pneumococcal vaccination status of study subjects is not available. Authors have mentioned rate of childhood pneumococcal vaccinations in Denmark. However, it will be relevant to discuss the incidence of pneumococcal vaccinations in adults. 
  4. It will be useful if the authors can comment on whether it were only a handful of subjects who developed infections repeatedly.
  5. I do not understand the following sentences:
    1. "During follow-up, 24 (2%) were re-transplanted and 169 (14%) died at median 223 (IQR 94 - 667) and 445 (IQR 170 - 938) days post-transplantation, respectively." Does this mean re-transplanted died at median 223 days? This sentence needs to be rephrased.  
    2. "---both the overall IR of S. pneumoniae and H. Influenzae infections and IR of invasive 196 infections were significant."--significant in comparison to what? 

Round 2

Reviewer 1 Report

Thank you for having addressed my suggestions.

Author Response

Thank you for your time.